# Validation of the Patient-Centred Care Competency Scale Instrument for Finnish Nurses

**DOI:** 10.3390/jpm11060583

**Published:** 2021-06-21

**Authors:** Riitta Suhonen, Katja Lahtinen, Minna Stolt, Miko Pasanen, Terhi Lemetti

**Affiliations:** 1Department of Nursing Science, University of Turku, 20014 Turku, Finland; katlahz@utu.fi (K.L.); minna.stolt@utu.fi (M.S.); misapas@utu.fi (M.P.); terhi.lemetti@hus.fi (T.L.); 2Turku University Hospital, 20014 Turku, Finland; 3City of Turku, Welfare Division, 20014 Turku, Finland; 4City of Helsinki, Department of Social and Health Care, 00099 Helsinki, Finland; 5University Hospital, 00029 Helsinki, Finland

**Keywords:** patient-centred care, competence, assessment, instrument, measurement, validity, reliability

## Abstract

Patient-centredness in care is a core healthcare value and an effective healthcare delivery design requiring specific nurse competences. The aim of this study was to assess (1) the reliability, validity, and sensitivity of the Finnish version of the Patient-centred Care Competency (PCC) scale and (2) Finnish nurses’ self-assessed level of patient-centred care competency. The PCC was translated to Finnish (PCC-Fin) before data collection and analyses: descriptive statistics; Cronbach’s alpha coefficients; item analysis; exploratory and confirmatory factor analyses; inter-scale correlational analysis; and sensitivity. Cronbach’s alpha coefficients were acceptable, high for the total scale, and satisfactory for the four sub-scales. Item analysis supported the internal homogeneity of the items-to-total and inter-items within the sub-scales. Explorative factor analysis suggested a three-factor solution, but the confirmatory factor analysis confirmed the four-factor structure (Tucker–Lewis index (TLI) 0.92, goodness-of-fit index (GFI) 0.99, root mean square error of approximation (RMSEA) 0.065, standardized root mean square residual (SRMR) 0.045) with 61.2% explained variance. Analysis of the secondary data detected no differences in nurses’ self-evaluations of contextual competence, so the inter-scale correlations were high. The PCC-Fin was found to be a reliable and valid instrument for the measurement of nurses’ patient-centred care competence. Rasch model analysis would provide some further information about the item level functioning within the instrument.

## 1. Introduction

Patient-centeredness in care has been reported to be a core health care value [1,2], and the optimal design for the delivery of healthcare [3,4] requiring specific competences from healthcare professionals [5,6]. Patient-centred care has also been found to be a healthcare core competency [7]. This core competency describes the professionals’ ability to identify patient expectations, preferences, and values, facilitating joint decision making, and acting with individual patients to deliver safe, effective, and compassionate care [8]. Patient-centred care is, therefore, a care approach which considers individual patient’s specific care needs [4,9,10] and is regarded as patients’ preferred care delivery process [11,12]. Importantly, patient-centeredness in care has been used as an attribute and indicator of quality [2,13] and patient safety [6], especially in the interaction between the patient and care provider [14].

The concept of patient-centred care is multidimensional, and includes domains at the individual, human level and at organizational levels [2,4,15,16,17]. The concepts of patient-centred care and person-centred care have been used interchangeably in the literature, e.g., [18], and defined in many ways. However, regardless of the particular title, the concepts include very similar core elements, with elements specific to professional groups in healthcare. Scholl and colleagues [18] (p. 1), in their model of patient-centredness in professionals and healthcare, identified 15 dimensions: “essential characteristics of the clinician; the clinician-patient relationship; clinician-patient communication; the patient as a unique person; the biopsychosocial perspective; patient information; patient involvement in care; involvement of family and friends; patient empowerment; physical support; emotional support; the integration of medical and non-medical care; teamwork and teambuilding; access to care; and the coordination and continuity of care”.

The difference between the concepts may be the intent of the focus on or viewpoint to ill-health issues or patient as a person, or the balance on these. However, the approach, typical in nursing is bringing the person to the centre of care, seeing the individual, trying to maintain their personhood, and valuing their personal experiences of life in relations with others in their social environment [19,20].

Competence has typically been defined in terms of “complex combinations of knowledge, skills, performance, attitudes and values” [20,21]. However, there seems to be no consensus about the meaning of these terms within the nursing profession [21,22,23]. Demonstrating this complexity further, both subjective (self-assessment) and objective (knowledge test and simulations) evaluations of competence have been used to assess competence. This complexity and lack of consensus is not surprising as assessing competence is many-sided, and rarely comprehensive [24].

Competence has been assessed and measured as a behavioural objective, within a psychological construct which includes decision-making and justifications for actions. In these assessments the self-assessing individuals justify their preferences and activities against defined, predetermined criteria. Measuring competence using self-assessment and standard instruments in this behavioural context has been criticized as reductionism, simplifying the assessed issues, dividing the complex processes of healthcare activities into single tasks, or sets of tasks, activities, and targets [24]. Helpfully, competence has also been divided into professional competence [25,26] and clinical competence, e.g., [27,28]. In a similarly helpful way, patient-centred care has also been separated into care competence needed by all professionals and specific competence, being context specific [29] and possibly varying between individual professionals [26]. As with general competence, the patient-centered care competence has been found to be multidimensional and can be conceptualised and operationalised in multiple ways [6].

Although the concept of patient-centred care has been used for many years and is considered widely to be fundamental for individualised healthcare [30,31], the competence requirements for such care provision remain largely undefined. Although empirical studies about patient-centred care competence are limited [6] these studies have pointed out the clear need for a special competence required for the provision of patient-centred care [5,6]. It follows that the assessment of patient-centred care competence forms the basis of support for the development and enhancement of patient-centred clinical care provision by nurses for patients [6]. The Patient-centred Care Competency (PCC) instrument was developed for measuring professional nurses’ competence in the delivery of patient-centred care in hospitals [6]. Within the PCC instrument, patient-centred care competence is defined in terms of knowledge, skills, and attitudes [6] (p. 45) and applies components from the Quality and Safety Education for Nurses (QSEN) faculty study [32]. The QSEN study [32] included the following components of competence for nurses: patient-centred care, advance event management, contributions to patient safety culture, effective communication, optimisation of human and environmental factors, risk management, and teamwork. The results of this study and the work of Hwang [6], formulating the PCC instrument, suggests that assessment could be based around the use of core competencies for nursing professionals including patient-centred care, as defined in terms of earlier literature, e.g., [11,19,33,34,35,36]. Thus, the PCC instrument includes assessment of the knowledge, skills and attitudes required in terms of respecting patients’ perspectives; promoting patient involvement in care processes; providing for patient comfort; and advocating for patients [6].

The aim of this current study was to assess (1) the reliability, validity, and sensitivity of the Finnish version of the PCC scale and (2) the self-assessed level of patient-centred care competency in two samples of Finnish nurses.

## 2. Materials and Methods

### 2.1. Design, Setting and Sampling

The assessment of the PCC uses the data from two earlier studies as secondary data where the PCC was used. These two studies employed cross-sectional survey designs and used the PCC instrument [6] (with permission from Hwang, also from Elsevier) as part of them. These separate datasets, labelled Dataset 1 and Dataset 2, were used to analyse the psychometric properties of the PCC, not fully reported previously.

Dataset 1 was collected electronically, between October 2016 and January 2017, from registered nurses working in one major university hospital in Southern Finland using self-administered questionnaires. Digitally delivered questionnaires were used in one organisation, with the assistance of the member of the organisation. Nurses were recruited with the help of nurse managers, who emailed the information letter and a link to the survey to potential participants. Inclusion criteria for participation were that respondents would: (1) be a registered nurse (RN) and (2) work in an acute hospital in-patient unit that cares for older patients. The researcher, together with the nurse managers, identified the units where older people form most of the patients cared for in the unit, including internal medicine units. In total 770 invitations were sent to a convenience sample of RNs working in 14 in-patient units, 223 nurses responded giving the response rate of 29%. The respondents’ mean age was 38.9 (standard deviation (SD) 11.6, range 23–64) and most of the respondents were female (n = 206). The highest level of nurse education was registered nurse (college or baccalaureate, n = 210). A few of the respondents had a master’s degree in nursing (n = 4) or had taken postgraduate classes in nursing (n = 9). The mean length of the work experience in the nurses’ current employment was 7.6 years (SD 7.3, range 0.1–35).

Dataset 2 was collected in written form, between October 2017 and June 2018 from nurses working in a University hospital and in primary health care units in Southern Finland. Paper-pencil format was the only useful technique as many organisations participated and access to study sites was demanding due to many participating organisations. A cluster sampling approach was used to recruit nurses from one university hospital (hospital) and two major cities (primary health care) offering care for older people in Finland. Respondents were informed about the study verbally and in writing. If the respondents were interested in participating in the study, they completed the paper questionnaire by hand, sealed it in an envelope and sent it to the researcher by mail. A total of 1435 questionnaires were delivered to potential respondents from 41 units in the university hospital and 19 units in primary health care: health centres: (n = 13); long-term care units (n = 2); home care (n = 2); and city hospitals (n = 2).

A total of 443 completed questionnaires (response rate 30.9%) were returned by hospital nurses (n = 240) and primary health care nurses (n = 203). The respondents mean age was 43.2 (SD 11.2, range 22–67) and most of the respondents were female (n = 426). The highest level of nurse education was that of Registered Nurse (college or polytechnic, n = 385). However, some of respondents’ highest level of nurse education was master of healthcare (polytechnic, n = 14), bachelor of healthcare (university, n = 19) or Master of healthcare (university, n = 14). The mean length of the work experience in their current unit was for nurses working in the university hospital, 8.6 years (SD 9.0, range 0.6–37.2) and for nurses working in primary health care, 6.2 (SD 6.3, range 0.8–32.0).

### 2.2. Instrument

The PCC scale [6] was used in both Finnish studies. The PCC was originally developed in Korea for the measurement of patient-centred competence of nurses in hospital settings. The competence was defined in terms of knowledge, skills and attitudes related to patient-centred care. The PCC consists of four sub-scales including Respecting patients’ perspectives (6 items), Promoting patient involvement in care processes (5 items), Providing for patient comfort (3 items) and Advocating for patients (3 items). The instrument uses a five-point Likert scale (1 minimal, 2 below average, 3 average, 4 good, 5 excellent) which the respondents used to rate their competencies. (subjective, self-assessment).

The 41 PCC items were developed, based on earlier literature and the QSEN faculty framework of competence in patient-centred care [32]. The items were then analyzed by experts who reduced the preliminary 41 items, to 25 after using the Content Validity Index (CVI) and cut off criterion of 0.70, and psychometric testing [6]. The expert panel members were eight members of the board of directors (Korean Quality Improvement Nurses Society) and three nursing professors, and they assessed the relevance of the proposed 41 items. The first version of the PCC was pretested in the sample of two head nurses and two registered nurses to verify the comprehensibility and clarity of the items. The PCC was then tested with a sample of 577 hospital nurses. Explorative factor analysis supported a four-factor solution explaining 64.7% of the variance and suggesting the final 17 items for the PCC [6]. The overall Cronbach’s alpha coefficients for the 17-item instrument was 0.92, and for the sub-scales, 0.80–0.85. Concurrent validity was assessed using the single item on patient-centred care performance using VAS (0–100), with Pearson’s correlation coefficient of 0.60 (*p* < 0.001).

### 2.3. Translation of the Instrument

The PCC with 17 items created by Hwang [6] was used, and the English language items were used. A standard forward-back translation method [37] was used to translate the PCC from the English version to the Finnish version by an official translator, whose work was analysed by two researchers, mainly interested in ensuring nursing terminology was well matched. Secondly, the translated version was back-translated into English by another official translator. Thirdly, the back translated version was compared with the original English language instrument. Finally, all three instrument versions were analysed at the same time by two experienced researchers. The final Finnish version included some minor changes of terms (professional terminology and most suitable term from two suggested) and the deletion of some redundant words. As languages differ from each other, the semantic, cultural, conceptual equivalence, and linguistic terminology were ensured [37].

### 2.4. Data Analysis

Data were analysed statistically using the SPSS for Windows (IBM SPSS, Chicago, IL, USA) version 22.0/24.0 and lavaan (0.6–7) package from R (version 4.0.2) statistical software. Firstly, descriptive statistics were calculated which described the two samples and study variables at the total scale and sum-variable level for the four sub-scales. The sum-variables were formed based on the theoretical background of the original instrument, see [6]. Secondly, the internal consistency reliability was examined using the Cronbach’s alpha coefficients and item analysis in both data sets (criterion ≥ 0.70) [38,39]. Item analysis included item-to-total correlations (criterion ≥ 0.30) and the percentage of the appropriate inter-item correlations (criterion 0.30 ≤ r ≤ 0.70). Thirdly, an explorative factor analysis (EFA) was conducted in the sample of 233 respondents (Dataset I). The Kaiser–Meyer–Olkin measure of sampling adequacy was 0.895, (acceptable value > 0.5) [40] and Bartlett’s test of sphericity (<0.001; where a *p* value less than 0.05 indicates that a factor analysis may be useful with the data) [41]. These two measures were used to assess the preconditions for factor analysis. An EFA, with principal axis factoring as the extraction method, and Promax with Kaiser Normalization for rotation method was computed. Fourthly, a confirmatory factor analysis (CFA) with maximum likelihood estimation was used to investigate the conceptualised four-dimensional structure (KMO = 0.92, Bartlett’s test < 0.001). Several indices with criteria were used to examine the goodness-of-fit of the model with Dataset 2: goodness-of-fit index (GFI); adjusted goodness-of-fit index (AGFI); Tucker–Lewis index (TLI) and the comparative fit index (CFI) (criterion >0.90 threshold for all mentioned fit indices [42,43], root mean square error of approximation (RMSEA <0.08) [42], and standardized root mean square residual (SRMR <0.08) [42]. Fifthly, to examine whether the PCC sub-scales measure distinct dimensions, inter-scale correlations between the PCC sub-scales were analysed. Finally, analysis of the sensitivity of the PCC was assessed comparing the PCC total and sub-scale scores within the two different facilities (contrasting groups), university hospital care (n = 233) and primary health care (n = 201) in Dataset 2 (n = 434). The distributions were non-normal and therefore, the Mann–Whitney U-test was used.

### 2.5. Ethical Considerations

Permissions for data collection were obtained from the participating organisations according to their specific ethical procedures. The studies using Dataset 1 and Dataset 2 were approved by the Ethics Committee of the University (34/2016/6 June 2016 and 4/2016/15 February 2016 respectively). Permission to use the PCC instrument was granted by the developer Jee-In Hwang (email) and Elsevier (reprint of the items). The respondents gave their voluntary informed consent for the studies by completing the questionnaires sent to them and posting them to the researcher, after having all information about the study in an introductory letter. Respondents were informed that they could withdraw from the study at any time.

## 3. Results

### 3.1. Descriptive Statistics on the Patient-Centred Care Competency (PCC) and Sub-Scales

In total, patient-centred care competence in Dataset I was rated in the ‘good’, level 4.04 (SD 0.46) and was higher compared to Dataset 2 (3.90, SD 0.42). At the sub-scale level (dataset 2 in parenthesis), competence providing for patient comfort was evaluated the highest in both data sets 4.31, SD 0.56 (4.13, SD 0.55) at the ‘good’ level. Competence in respecting patients’ perspectives 4.12 SD 0.49 (3.99, SD 0.42) and advocating for patients 4.03 SD 0.56 (3.84, SD 0.56) was also rated ‘good’. Competence promoting patient involvement in care processes was assessed the lowest at 3.78 SD 0.56 (3.70, SD 0.49).

### 3.2. Psychometric Properties of the PCC

#### 3.2.1. Internal Consistency Reliability

The internal consistency measured using Cronbach’s alpha coefficients (Dataset 2 in parenthesis) was α = 0.93 (0.91) for the total scale and ranged from α = 0.78 (0.74) to α = 0.85 (0.83) for the sub-scales (Table 1). Item analysis (Dataset I) provided some evidence about the internal consistency of the items within the total scale and its four sub-scales. All items were closely tied to its construct as all item-to-total correlations reached an acceptable level (≥0.40). Correlations between the items in the given sub-scale (inter-item correlations) were acceptable in three sub-scales (PCC2–4), as was 87% of the inter-item correlations within the PCC total and the sub-scale PCC1 (Table 1).

#### 3.2.2. Construct Validity

Firstly, the exploratory factor analysis of Dataset I suggested a three-factor solution based on scree plot and eigen values (criterion eigen value ≥ 1). The Pattern matrix showed a clear structure for three sub-scales (Factors 3, 2 and 1) which explained 57.7% of the variance. However, the fourth sub-scale was not independent as the items loaded on the first sub-scale (Factor 1, item 15) and the third sub-scale (Factor 3, item 16 and item 17). A four-factor model was also examined, with a 61.2% explained variance, and an eigen value range of 8.04–0.87, and a last factor with an eigen value of less than 1.0. (Table 2).

Secondly, a Confirmatory Factor Analysis was conducted using the Dataset 2 (n = 421 valid cases). The Chi-square test did not show a model fit (χ^2^ = 283.70, df = 113, *p* < 0.01) possibly due to the large sample size. Other goodness-of-fit indices were as follows: CFI 0.93, TLI 0.92, GFI 0.99 and adjusted AGFI 0.99, all at an acceptable level (criterion >0.90). The RMSEA was 0.065 (criterion <0.08) and the SRMR of 0.045 (criterion <0.08) was also a good fit to the model. The Item loadings ranged were: PCC1 (0.584–0.729), PCC2 (0.612–0.681), PCC3 (0.726–0.824) and PCC4 (0.699–0.707), all of which were acceptable (criterion >0.40) (Figure 1).

The average variance extracted (AVE), the amount of variance captured by a construct in relation to the amount of variance due to measurement error) [44] was 0.44 (PCC1), 0.41 (PCC2), 0.49 (PCC3) and 0.47 (PCC4) implying that the discriminant validity may not be ideal but includes variance based on measurement bias. However, the factor loadings were as follows, PCC1 (0.584–0.729, composite reliability (CR) 0.82), PCC2 (0.612–0.681, CR 0.77), PCC3 (0.726–0.824, CR 0.83) and PCC4 (0.699–0.707, 0.82), some being lower than 0.70.

#### 3.2.3. Inter Scale Correlations

The correlations between the PCC sub-scales ranged from 0.57 to 0.88 showing the scales are related to each other and to the latent variable, PSS (Figure 1).

### 3.3. Sensitivity to Context/Contrasting Groups

The competence assessments by nurses working in specialised hospital care and those working in primary health care were not statistically significantly different in the total scale or any of the sub-scales (total PCC, Mann–Whitney U =20,804, *p* = 0.659), PCC1 (U = 22,484, *p* = 0.628), PCC2 (U = 21,763, *p* = 0.519), PCC3 (U = 23,620, *p* = 0.745) or PCC4 (U = 20,110, *p* = 0.980).

## 4. Discussion

Two datasets (1 and 2) were used to validate the PCC instrument in the Finnish healthcare context. The psychometric properties computed suggest acceptable reliability, content and construct validity and sensitivity. The PCC was developed and tested in the hospital environment in Korea by Hwang [6] to measure patient-centred care competency, and the findings in this current study are similar to those of Hwang [6]. The purpose of the PCC instrument is clearly defined, has meaningful content in the Finnish context and has acceptable internal consistency, face validity and construct validity [45]. These results demonstrate that the instrument is a useful measure of patient-centred care competency. In the Finnish studies the nurses assessed their overall patient-centred care competency higher 4.04 (Dataset I), 3.90 (Dataset 2) compared to the Korean studies, 3.58 [6] and 3.61 [46] respectively. The sub-scale “providing for patient comfort’’ was assessed the highest and the sub-scale “promoting patient involvement’’ was assessed the lowest in all four studies, increasing the inter-rater reliability of the instrument.

This current study also shows that the instrument is suitable for use in different health care environments, for example, home care and long-term care units (Dataset 2) as the findings are comparable between Dataset I and Dataset 2. However, because the data from home care and long-term care units, Dataset 2, is limited, and the staffing profiles differ from those in hospital environments, more research is needed to be more certain of this.

The sub-scales contain items related to general patient relations, patient involvement, safety, and teamwork [6,46] which are shared goals of all health care workers, for example clinicians and practical nurses. Therefore, it can be argued that although the instrument was originally developed for use in nursing studies, it could be used, with minor adaptations, to measure the patient-centred care competency of other healthcare workers such as physiotherapists and public health nurses.

Translating instruments for nursing studies is necessary [47] for researchers to have access to the many valid and reliable instruments available [37]. The standard forward-back translation method [37] for the Finnish version of the PCC achieved equivalence [47] with the original English and also Korean version. The Likert-type response options of the PCC are sensitive, and the instrument is comprehensible to the healthcare respondents as it suits their demographic and educational backgrounds [48]. The PCC is easy to complete taking a short time only, making it a useful research tool [48]. However, more research is needed to further validate the instrument, for example in diverse healthcare environments and among different professionals working in health care.

Patient-centred care competence is a multidimensional concept which can be defined and measured from different perspectives and multiple ways. This study provides more information about the validation of one instrument, the PCC instrument [6], used for measuring nurses’ patient-centred care competence. The results of this study suggest that the PCC instrument is a valid, reliable, and sensitive instrument that could be used to measure Finnish nurses’ patient-centred care competence and may be adapted to measure the patient-centred competence of other health care workers. Cowan et al. [24] suggests that the self-assessed instrument oversimplifies a complex process, applying also to PCC. However, this instrument provides important information about nurses’ own views of their patient-centred care competence. Combining this information with other measurements, for example, peer, co-workers’ and patients’ perspectives, can provide a more comprehensive understanding of patient-centred care competence in the workplace.

### Methodological Considerations, Validity and Limitations

Some of the methodological considerations of the study, relate to data collection, sample size and instrumentation, and warrant further discussion. The data in this study were taken from two independent studies, using same instrument, the PCC [6]. Due to the secondary nature of the data, it was not possible to increase the sample size. However, the recommended sample size for a study like this, is at least five respondents per item in cross-sectional survey studies [38,39] which was realized in both studies. The PCC was translated to Finnish following the recommended forward-back translation process and semantic and linguistic equivalence were confirmed. As the PCC is a self-assessment instrument, the results may be affected by social desirability bias [49], as the participants may have evaluated their competence higher than in reality. This is to be expected as patient-centeredness in care, strongly rooted in the value-based healthcare, and often discussed in policy and strategic documents is a pillar of professional nurse education. There is no method available to evaluate competence in such value-laden activities, including the knowledge, skills, and attitudes objectively [21,24]. The PCC items self-assess knowledge, skills, and attitudes of the practicing nurse. The next step could be the development of a more objective assessment method. However, it might be very challenging objectively to measure all the aforementioned dimensions of competence. This may possibly lead to behavioural level assessment. It has been criticized that the competence of healthcare professionals is multidimensional, demanding several assessment methods.

Cronbach’s alpha coefficients, indicating the level of internal consistency homogeneity of the total scale, in both datasets were (0.91–0.93). This high value suggests some scale items have similar meaning, requiring item deletion. However, the Cronbach’s alpha coefficients are lower and more acceptable at the sub-scale level in both datasets (0.78–0.85 and 0.74–0.83). The correlation coefficients (also in the CFA) between the factors, especially between PCC4 and the other factors PCC1, PCC2 and PCC3, are high, suggesting there may be some overlap within the scales. The chi-square statistics did not show a model fit with a *p* value less than 0.05. However, this statistic is known to be sensitive to sample size [50] and is, therefore, insufficient in this study for the assessment of goodness-of-fit. The other goodness-of-fit indices were all acceptable.

The average variance extracted (AVE 0.41–0.49 within the sub-scales) implied that the discriminant validity may not be ideal and includes variance based on measurement bias. The factor loadings were mostly acceptable PCC1 (0.584–0.729), composite reliability CR 0.82), PCC2 (0.612–0.681, CR 0.77), PCC3 (0.726–0.824, CR 0.83) and PCC4 (0.699–0.707, 0.82). The criteria given (0.5 > AVE > 0.4) [44] if the composite reliability CR >0.6, is acceptable (here all CR > 0.7), but may suggest revision for the total scale, by deleting some possibly redundant items. The PCC scale could benefit from a Rasch analysis which would further analyse item-level reliability and provide information about how participants response patterns relate to the difficulty of the items. For the known-group validity testing, the findings regarding the total scale or its sub-scales indicate that competence was assessed in the hospital care context and in primary health care context in similar ways, even though there was a variety of service provision.

## 5. Conclusions

Patient-centred care has a central role in healthcare worldwide [30,31] and has been found to improve health literacy and patient engagement, be effective and cost-effective [31] and can be used as an important indicator of care quality [2,13] and patient safety [6]. In nursing, patient-centred care competence has been identified as a core competency for nurses [7,32] and is needed to guide the development of care. The PCC instrument measures nurses’ patient-centred care competence in terms of knowledge, skills, and attitudes through self-assessment. In this current study, the PCC was validated in the Finnish healthcare system by registered nurses using self-assessment. The PCC has proven reliability, construct validity and sensitivity for the measurement of non-contextual, specific competence. Further research into the analysis of item-level discrimination with, for example, Rasch modelling, will identify any overlapping items and person fit in evaluations, including the identification of nurse-related characteristics. The results of this study showed there is some room for improvement in the promotion of patient involvement in their care.

## Figures and Tables

**Figure 1 jpm-11-00583-f001:**
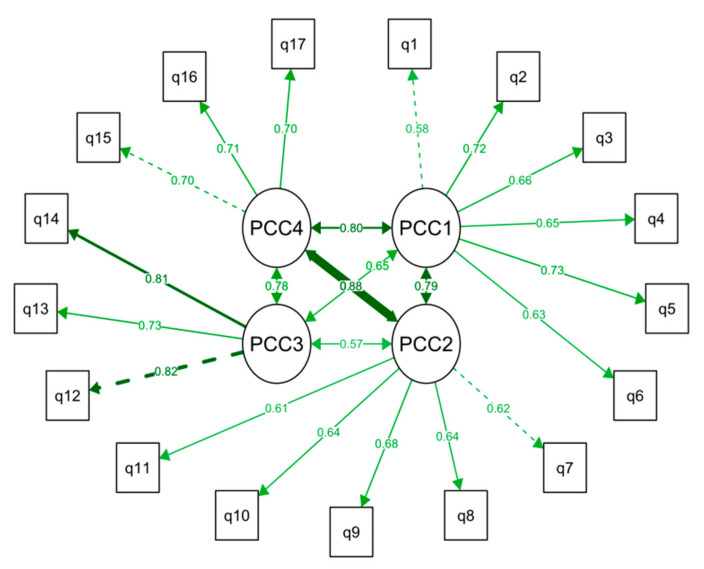
Construct validity of the Patient-centred Care (PCC) instrument based on confirmatory factor analysis (CFA) and inter-scale correlations PCC1–PCC4. q refers to items, numbers refer to factor loadings within each sub-scale. Model fit statistics: comparative fit index (CFI) 0.93, Tucker–Lewis index (TLI) 0.92, goodness-of-fit index (GFI) 0.99, adjusted goodness-of-fit index (AGFI) 0.99, root mean square error of approximation (RMSEA) 0.065, standardized root mean square residual (SRMR) 0.045.

**Table 1 jpm-11-00583-t001:** Descriptive statistics of the Patient-centred Care Competency (PCC) total and sub-scales, Cronbach’s alpha and item analysis.

Scale	n of Items	n	Mean (SD)	α *	Item-to Total r	Average Inter-Item r
					Range (% Criteria) §	(Range), % Criteria #
Dataset 1						
PCC total	17	223	4.04 (0.46)	0.93	0.518–0.704 (100)	0.437 (0.035–0.70) 87%
Respecting patients’ perspectives	6	223	4.13 (0.49)	0.84	0.444–0.681 (100%)	0.472 (0.243–0.637) 87%
Promoting patient involvement in care processes	5	223	3.78 (0.56)	0.85	0.571–0.705 (100%)	0.531 (0.414–0.628) 100%
Providing for patient comfort	3	223	4.31 (0.56)	0.85	0.686–0.743 (100%)	0.655 (0.619–0.700) 100%
Advocating for patients	3	223	4.03 (0.56)	0.78	0.552–0.679 (100%)	0.537 (0.456–0.613) 100%
Dataset 2						Inter-item range (5) #
PCC total	17	434	3.90 (0.42)	0.91	0.51–0.64 (100%)	0.28–0.54 (92%)
Respecting patients’ perspectives	6	434	3.99 (0.44)	0.82	0.52–0.67 (100%)	0.31–0.60 (100%)
Promoting patient involvement in care processes	5	434	3.70 (0.49)	0.77	0.50–0.58 (100%)	0.31–0.48 (100%)
Providing for patient comfort	3	433	4.13 (0.55)	0.83	0.60–0.74 (100%)	0.55–0.69 (100%)
Advocating for patients	3	431	3.84 (0.56)	0.74	0.52–0.61 (100%)	0.43–0.55 (100%)

* α Cronbach’s alpha coefficient. § item to total correlation r > 0.3. # inter item correlation 0.30 < r < 0.70.

**Table 2 jpm-11-00583-t002:** Exploratory factor analysis (EFA) Pattern matrix, Dataset I (n = 223), items of competence in the PCC.

Item	Respecting Patients’ Perspectives	Communality	Factor 1	Factor 2	Factor 3
Item 1	Value seeing health-care situations through patients’ eyes	0.587	−0.002	0.160	**0.604**
Item 2	Elicit patient values, preferences and needs as part of clinical interview, implementation of care plan, and evaluation of care	0.617	−0.167	0.421	**0.509**
Item 3	Integrate understanding of multiple dimensions of patient-centred care such as patient and family preferences	0.633	−0.102	0.320	**0.555**
Item 4	Communicate patient values, preferences and need to other health-care team members	0.603	−0.144	0.142	**0.686**
Item 5	Provide patient-centred care with sensitivity and respect for the diversity of human experience	0.640	0.210	−0.334	**0.902**
Item 6	Support patient-centred care for individuals and groups whose values differ from own	0.483	0.276	0.003	**0.389**
	**Promoting Patient Involvement in Care Processes**				
Item 7	Examine barriers to active involvement of patients in the care processes	0.593	−0.002	**0.430**	0.386
Item 8	Assess level of patient’s decisional conflict and provide access to resources	0.611	0.060	**0.641**	0.159
Item 9	Describe strategies to empower patients or families in all aspects of care process	0.623	−0.156	**0.897**	−0.017
Item 10	Engage patients or designated surrogates in active partnerships that promote health, safety and well-being, and self-care management	0.578	0.221	**0.775**	−0.195
Item 11	Respect patient preferences for degree of active engagement in care process	0.574	0.281	**0.310**	0.231
	**Providing for Patient Comfort**				
Item 12	Assess presence and extent of pain and suffering	0.631	**0.855**	−0.042	−0.061
Item 13	Assess levels of physical and emotional comfort	0.627	**0.914**	−0.157	−0.003
Item 14	Elicit expectations of patient and family for relief pain, discomfort and suffering	0.698	**0.709**	0.230	−0.062
	**Advocate for Patients**				
Item 15	Facilitate informed patient consent for care	0.659	0.505	0.165	0.201
Item 16	Communicate care provided and needed at each transition in care	0.608	0.303	0.108	0.424
Item 17	Participate in building consensus or resolving conflict in the context of patient care	0.533	0.231	0.191	0.398
	Eigen value		8.041	1.924	0.996
	% of explanation		44.8	9.3	3.6
	Cumulative % of explanation		44.8	54.057	57.7

Extraction Method: principal axis factoring; Rotation method: Promax with Kaiser normalization. Bold: representing original structure. Items reprinted from *Nursing Outlook* 55(1), Cronenwett et al. Quality and safety education for nurses 122–131, Table 1. Copyright (2007), with permission from Elsevier.

## Data Availability

No new data were created or analyzed in this study. Data sharing is not applicable to this article.

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
