# Peer review of "Validation of the Patient-Centred Care Competency Scale Instrument for Finnish Nurses"

_jpm, 2021, doi:10.3390/jpm11060583_

Round 1

Reviewer 1 Report

An highly detailed validation study of the Patien-centred Care Competency Scale, expertly carried out, with comprehensive analysis and discussion of validity and limitations.

Author Response

We kindly thank the editorial team and the reviewers for the critical comments raised. All have been considered carefully, and taken into account in revising the manuscript. Some of the comments raised were answered in the revision notes, and the corrections done in track changes action.

Reviewer 2 Report

Thank you for the opportunity to review this paper.  I appreciate that it takes a lot of effort to prepare work for publication and I hope that you find my comments helpful.

Introduction

Excellent review and integration of the literature to support what constitutes patient centred care. The complexities associated with what constitutes competency and the difficulties associated with measuring competency are well articulated.

Materials and Methods   

Dataset 1 was completed electronically and Dataset 2 was collected in written form; why was this decision taken? Please provide a rationale.

The explanation of how the instrument was reduced from 41 to 17 items warrants further discussion; for example, how many times was it tested?  Who were the experts involved in making this decision? Line 163

It would be helpful to include the actual 17 item tool used in the study.

The final Finnish version included some minor 179 changes of terms and the deletion of some redundant words”. This requires further explanation please as to how this occurred.

There is no reference to a pilot study or any test for face validity of the final Finnish version of the PCC instrument.

The instrument generates subjective responses from participants; to assess competency there is the need to have an external independent assessor.  For this reason, it is difficult to know how the findings can be validated. It is acknowledged that the author has considered this under limitations of the study.  

Although I am not an expert in statistical analysis I found the explanation of the statistics to be clear and the tests appear to be appropriate.  However, there is no reference to the qualitative data generated in responses to, for example, item 9 “Describe strategies to empower patients or families in all aspects of care process”

Where was the qualitative free response data captured? How was it analysed?

Given that the instrument was to measure competency and it was completed by the individual nurse with no external assessor it is difficult to evaluate how relevant the findings are.

The conclusion is well written offering a summary of the key findings and relevance to health professional education.

Summary:

 The research will contribute to future developments in this important area and should be of interest to the JPM reader. However the concerns raised require to be addressed before consideration can be given to publication . Please proof read the submission to ensure that there are no omissions for example :  Funding: This research was funded by The Turku University Hospital, special grant-in aid VTR, 396 g_r_a_n_t_ _n_u_m_b_e_r_ _1_3_2_3_8_” _a_n_d_ _“T_h_e_ _A_P_C_ _w_a_s_ _f_u_n_d_e_d_ _b_y_ _X_X_X_ _

Author Response

(The authors gave the same response as above.)

Round 2

Reviewer 2 Report

Thank you.  I am satisfied that all the points raised in my previous review have been adequately addressed.  I appreciate the detailed responses from the authors .

7th June 2021